# Glucose- and Non-Glucose-Induced Mitochondrial Dysfunction in Diabetic Kidney Disease

**DOI:** 10.3390/biom12030351

**Published:** 2022-02-23

**Authors:** Marie Ito, Margaret Zvido Gurumani, Sandra Merscher, Alessia Fornoni

**Affiliations:** Department of Medicine, Katz Family Division of Nephrology and Hypertension, Peggy and Harold Katz Family Drug Discovery Center, Miller School of Medicine, University of Miami, Miami, FL 33136, USA; mxi288@miami.edu (M.I.); mzg13@miami.edu (M.Z.G.)

**Keywords:** diabetic kidney disease, mitochondrial dysfunction, mitochondrial reactive oxygen species, Warburg effect

## Abstract

Mitochondrial dysfunction plays an important role in the pathogenesis and progression of diabetic kidney disease (DKD). In this review, we will discuss mitochondrial dysfunction observed in preclinical models of DKD as well as in clinical DKD with a focus on oxidative phosphorylation (OXPHOS), mitochondrial reactive oxygen species (mtROS), biogenesis, fission and fusion, mitophagy and urinary mitochondrial biomarkers. Both glucose- and non-glucose-induced mitochondrial dysfunction will be discussed. In terms of glucose-induced mitochondrial dysfunction, the energetic shift from OXPHOS to aerobic glycolysis, called the Warburg effect, occurs and the resulting toxic intermediates of glucose metabolism contribute to DKD-induced injury. In terms of non-glucose-induced mitochondrial dysfunction, we will review the roles of lipotoxicity, hypoxia and vasoactive pathways, including endothelin-1 (Edn1)/Edn1 receptor type A signaling pathways. Although the relative contribution of each of these pathways to DKD remains unclear, the goal of this review is to highlight the complexity of mitochondrial dysfunction in DKD and to discuss how markers of mitochondrial dysfunction could help us stratify patients at risk for DKD.

## 1. Introduction

The number of patients with diabetes mellitus is expanding, and diabetic kidney disease (DKD) is an important cause of diabetic microvascular complications that constitutes an independent risk factor of mortality and cardiovascular events [1].

The kidney is one of the most energy-demanding organs and, after the heart, has the second highest expression of proteins involved in mitochondrial function and oxygen consumption [2,3]. The kidney requires energy mainly for solute reabsorption, among other tasks including waste removal, maintenance of electrolyte and fluid balance and acid–base homeostasis [4]. The generation of an ion gradient across the plasma membrane by Na^+^/K^+^-ATPase is essential for solute reabsorption. Therefore, mitochondrial dysfunction is postulated to play a central role in the pathogenesis and progression of kidney diseases including DKD [5,6,7].

Since each component of the nephron has a distinct role and different energy requirement, the causes and phenotypes of mitochondrial dysfunction may differ among cell types in the kidney. In this review, we will discuss glucose-related and unrelated pathways of mitochondrial dysfunction contributing to DKD in terms of the categories as well as the causes.

## 2. Mitochondrial Dysfunction in DKD

Due to the critical role of mitochondria as the powerhouse of cells, mitochondrial dysfunction traditionally referred to an alteration in the production of adenosine triphosphate (ATP) by oxidative phosphorylation (OXPHOS). However, as our understanding of the various roles played by mitochondria has expanded, mitochondrial dysfunction now includes any abnormal biological process in mitochondria [7]. In this section, we will discuss different categories of mitochondrial dysfunction occurring in DKD and discuss their potential causative role (Figure 1).

### 2.1. Mitochondrial Oxidative Phosphorylation (OXPHOS)

As the powerhouse of cells, the central role of mitochondria is the production of adenosine triphosphate (ATP). Metabolites from glucose, lipids and amino acids are transported into the mitochondrial matrix, serving as substrates of the tricarboxylic acid (TCA) cycle. NADH and FADH_2_ are generated along with the reaction feed electrons into Complexes I and II of the electron transport chain (ETC). As electrons are transported through the ETC, H^+^ ions are pumped into the intermembrane space. Complex V or ATP synthase uses this proton gradient to generate ATP (Figure 2). The indices of OXPHOS activity and fitness include oxygen consumption rate (OCR), ATP production, membrane potential and the evaluation of each complex (activity, formation). Generally, it has been observed that OCR in the kidney cortex is increased in early DKD, followed by a decrease as DKD progresses, whereas in glomeruli and podocytes the OCR is decreased in both in early and late phases of the disease [5]. Although some discrepancy exists between studies, ATP production and complex activity have been demonstrated to be decreased at least in the late stage of DKD [8,9]. The contribution of decreased activation of OXPHOS to DKD can be inferred from the observation that some genetic mutations in OXPHOS, such as single-nucleotide polymorphisms (SNPs) in coenzyme Q5 (*COQ5*) and cytochrome *c* oxidase (*COX6A1*), are linked to DKD in humans [10]. *COQ5* encodes methyltransferase located in the mitochondrial matrix and *COX6A1* encodes a subunit of cytochrome c, which is part of the ETC.

### 2.2. Mitochondrial Reactive Oxygen Species (mtROS)

Ever since Brownlee and colleagues proposed that hyperglycemia-induced mitochondrial reactive oxygen species (mtROS) was the unifying mechanism of diabetic microvascular complications in 2000, this paradigm has been prevalent [11,12]. Recently, the source of ROS in DKD and the pathogenic role of ROS have become controversial [13,14]. Although there may be a consensus that ROS-induced damage is increased in DKD, conflicting studies exist regarding the change in mtROS production, which can be attributed to the different methods used to detect mtROS or the varying models or timepoints of DKD. In both live and fixed db/db mouse kidneys, increased mitochondrial ROS was observed using a mitochondrial matrix-localized reduction–oxidation-sensitive green fluorescent protein probe [15]. By contrast, in streptozotocin (STZ)-injected C57BL/6J mice and Ins2-Akita mice (DBA/B6 F1 mice), decreased mitochondrial superoxide was observed upon systemic administration of dihydroethidium (DHE) both in live and fixed kidneys [16]. The latter study does not preclude ROS production in other cell compartments, including the endoplasmic reticulum (ER) or enzyme systems such as nicotinamide adenine dinucleotide phosphate oxidase (Nox). Notably, the restoration of mitochondrial biogenesis and OXPHOS activity by adenosine monophosphate-activated protein kinase (AMPK) activation increased mtROS and ameliorated the DKD phenotype, arguing against the role of mtROS in inciting DKD.

ROS does not play an exclusively detrimental role in cell biology. Mitochondrial hormesis is the concept that slightly enhanced mitochondrial superoxide at baseline can decrease susceptibility to more severe cell stress [13]. ROS also plays an essential role in certain cell signaling pathways, requiring further elucidation of the intricate characteristics of ROS.

### 2.3. Biogenesis

Cells cope with increasing energy demand by increasing mitochondrial biogenesis, in which functional mitochondria are generated by duplication of mitochondrial DNA (mtDNA) and subsequent binary fission. Peroxisome proliferator-activated receptor γ coactivator 1α (PGC1α) plays a central role in mitochondrial biogenesis [17]. PGC1α is a transcriptional regulator of mitochondrial metabolic pathways such as oxidative phosphorylation (OXPHOS), the TCA cycle and fatty acid metabolism. PGC1 peroxisome proliferator-activated receptors (PPARs) and estrogen-related receptors (ERRs) can also serve as a coactivator of PGC1α. PGC1α dimerizes with transcriptional coactivators to regulate downstream gene transcription, and these partners include cyclic AMP-responsive element-binding protein (CREB) nuclear respiratory factors 1 and 2 (NRF1 and NRF2) and activated PPARs and ERRs [4]. Nutrient-sensing pathways like the mechanistic target of rapamycin (mTOR), AMPK, sirtuin, cyclic AMP (cAMP) and cyclic guanosine monophosphate (cGMP) regulate PGC1α directly or indirectly.

In DKD, although there is some discrepancy between studies possibly due to the analysis at different disease stages, PGC1α activity is considered to be increased in the early stage of diabetes, as demonstrated in 8-week-old db/db mice, followed by a decrease in activity at later stages, as demonstrated in pretransplant patients and mice 24 weeks after diabetes induction with STZ injection [8,16,18,19]. Taurine upregulated gene 1 (*Tug1*), a long noncoding gene, was described as a regulator of PGC1α in podocytes in DKD [20]. It was demonstrated that *Tug1* binds to an element upstream of *Ppargc1a* and interacts with PGC1α binding to its own promoter, subsequently enhancing *Ppargc1a* promoter activity.

### 2.4. Mitochondrial Fission and Fusion

Mitochondria are dynamic organelles which undergo tightly controlled processes of fission and fusion. Mitochondrial fission is mediated by dynamin-1-like protein (DRP1) and its receptors such as fission factor 1 (FIS1), mitochondrial fission factor (MFF) and mitochondrial dynamics proteins of 49 and 51 kDa (MID49 and MID51). Mitochondrial fusion is mediated by the long isoforms of optic atrophy protein 1 (OPA1), which plays a role in inner mitochondrial membrane fusion, and the mitofusins (MFN1 and MFN2) which play a role in outer mitochondrial membrane fusion [5,6].

Although the increase in mitochondrial fission and fusion factors such as the long isoforms of OPA1, MFN1, MFN2 and MFF were observed in early DKD, mitochondria were consistently fragmented throughout early and late stage in STZ-injected rats [8]. Human kidney biopsies of patients with DKD also demonstrated fragmented mitochondria in podocytes and proximal tubular cells [21,22]. Consistent with increased fission and decreased fusion, Drp1 and FIS1 expression was increased, while MFN2 expression was shown to be decreased in tubules in the latter study.

### 2.5. Mitophagy

Autophagy is a pathway that degrades and recycles damaged organelles and macromolecules, and selective autophagy of mitochondria is termed as mitophagy. Mitophagy has a critical role in the maintenance of mitochondrial quality by removing damaged mitochondria. Mitophagy can be mediated by the phosphatase and tensin homolog-induced putative kinase 1 (PINK1)/parkin-mediated pathway and other outer mitochondrial membrane proteins such as BCL2/adenovirus E1B 19 kDa protein-interacting protein 3 (BNIP3) and NIP3-like protein X (NIX), or the mitophagy receptor FUN14 domain-containing protein 1 (FUNDC1).

The PINK1 and parkin-mediated pathway has been more extensively investigated than the others [23]. PINK1 has a Ser/Thr kinase domain and is found inserted into both the inner and outer mitochondrial membranes. In healthy mitochondria, PINK1 is cleaved at two points by mitochondrial proteases, leading to its dissociation from the mitochondrial membrane and degradation by the ubiquitin-proteasome system. In depolarized mitochondria, PINK1 escapes cleavage and stably resides in the outer membrane. Subsequently, PINK1 homodimerizes and autophosphorylates to recruit E3 ubiquitin ligase parkin and ubiquitin, directing the mitochondria to the mitophagy pathway. In the ubiquitin-independent pathway, outer mitochondrial membrane proteins such as BNIP3, NIX or FUNDC1 recruit microtubule-associated protein 1A/1B light chain 3 (LC3) and induce mitophagy under certain stimuli including hypoxia [24,25]. Cardiolipin, which is located in the inner mitochondrial membrane under normal conditions, is externalized by certain stimuli and detected by LC3, facilitating the engulfment of the mitochondria by autophagosomes [26]. P62 is a marker of autophagy cargo, and its accumulation can indicate stagnation in degradation via autophagic flux. In general, basal mitophagy levels of podocytes are high, which can be attributed to their terminally differentiated characteristics. In contrast, in tubular cells the mitophagy level is low at baseline but it can be induced as a consequence of stress.

Mitophagy is suppressed in DKD, which was demonstrated by low PINK1/parkin-expression levels in podocytes of STZ-induced diabetic mice and increased p62 expression levels in tubular cells of biopsy obtained from patients with DKD [27,28,29]. Thioredoxin-interacting protein (TXNIP) was implicated in the suppression of tubular autophagy and mitophagy induced by high glucose [27]. High glucose was also shown to inhibit the transcriptional activity of forkhead-box class O1 (FoxO1) via its phosphorylation by Akt (protein kinase B), leading to the downregulation of PINK1 [29]. The protective effect of mitoquinone on DKD was partially attributed to the restoration of PINK1 and parkin protein expression in tubular cells via NRF2 activation [30].

### 2.6. Urinary Mitochondrial Biomarker

It is of value to detect DKD in the early disease stage in a reliable manner, preferentially before microalbuminuria or a decrease in the estimated glomerular filtration rate (eGFR) becomes evident, to prevent further progression to end stage kidney disease. Since mitochondrial dysfunction is thought to precede overt histological changes in DKD, biomarkers of mitochondrial dysfunction have recently been vigorously investigated. Even though those studies investigated patients with established DKD, they can be useful to identify biomarkers in patients affected by DKD. A study of metabolites in urine found global suppression in mitochondrial respiration in patients with DKD compared to controls without DKD [19]. In concordance with the notion that mtDNA is released from damaged tubular cells, the same study demonstrated increased mtDNA in urine. Although urinary mtDNA had a modest but significant inverse correlation with intra-renal mtDNA and eGFR at baseline, a positive correlation with interstitial fibrosis was found. Urinary mtDNA or mtDNA of biopsy specimens did not significantly correlate with eGFR decline in the 24 months of follow up [31]. Further investigations are warranted to determine if mitochondrial biomarkers can be used clinically.

## 3. Glucose-Induced Mitochondrial Dysfunction in DKD

In glomerular cells, glucose is taken up by glucose transporters (GLUTs) via facilitated diffusion transport. The expression pattern of each member of the GLUTs is cell-specific. Mesangial cells express GLUT1 and 4, podocytes express GLUT1, 4 and 8 and endothelial cells express GLUT1 [32]. In contrast, tubular cells reabsorb glucose from the glomerular filtrate mainly via sodium-dependent glucose cotransporters (SGLTs). Glucose reabsorbed by tubular cells is dissipated across GLUTs in the basolateral plasma membrane and diffuses into the interstitium. Proximal tubular cells can also produce glucose via gluconeogenesis, which is increased in type 2 diabetes [33]. Hyperglycemia is the major pathogenetic factor and some intermediate metabolites of glucose metabolism have also been implicated in contributing to cell injury in DKD. In the unifying hypothesis offered by Brownlee and colleagues in 2000, it was proposed that the overproduction of mtROS due to increased influx into OXPHOS activated the nuclear DNA-repair enzyme poly(ADP-ribose) polymerase (PARP), leading to the decreased activity of glyceraldehyde 3-phosphate dehydrogenase (GAPDH) and the subsequent accumulation of toxic intermediates of glucose metabolism [11]. However, later studies have suggested that mitochondria are dysfunctional and that other mechanisms, in addition to a mere increase in substrates, contribute to increased glycolysis and the subsequent accumulation of toxic metabolites.

### 3.1. Warburg Effect

In general, cancer cells display enhanced glycolysis and impaired oxidative phosphorylation. While the short-time and reversible shift of this metabolic process is called the Crabtree effect, the long-term metabolic reprogramming is called the Warburg effect [34]. The Warburg effect is the term originally used to describe a shift from OXPHOS to aerobic glycolysis (in which lactate is produced as a final product from glucose) in cancer [35]. Recent studies have demonstrated that this shift also takes place in diabetic kidneys [9,19]. Mitochondria in diabetic tissues are dysfunctional, as discussed above. Transcriptomic, metabolomic and metabolite flux analysis showed increased glucose metabolism and decreased mitochondrial function in the kidney cortices of db/db mice [9]. A metabolomics analysis of urine samples demonstrated significant decreases of 13 metabolites in patients with DKD compared to healthy controls, 12 of which were associated with mitochondrial metabolism [19]. Whether mitochondrial dysfunction causes the shift to glycolysis or if increased glycolysis causes mitochondrial dysfunction remains to be established [36].

Pyruvate kinase is the enzyme which catalyzes the conversion of phosphoenolpyruvate to pyruvate, the last and irreversible step of glycolysis. Decreased pyruvate kinase M2 activity was identified as the possible mechanism of the Warburg effect in DKD by Qi and colleagues [37]. They first conducted proteomics analysis of glomeruli isolated from patients with type 1 diabetes who did not develop DKD for over 50 years (protected) and those with a histological confirmation of DKD (unprotected). Some enzymes involved in glucose metabolism and antioxidation were found to be increased in protected glomeruli, and in particular, pyruvate kinase M2 (PKM2) expression and activity were upregulated. In mechanistic studies using high-glucose-treated podocytes and STZ-injected mice, it was confirmed that PKM2 activity was decreased in the DKD mouse model, that PKM downregulation contributed to DKD exacerbation and that pharmacological activation of PKM2 reversed the elevation in toxic glucose metabolites and mitochondrial dysfunction. Similarly, decrease in pyruvate kinase activation causes the Warburg effect in cancer [38].

Other factors that are postulated to induce the Warburg effect in DKD include sphingomyelin and fumarate accumulation [39]. Matrix-assisted laser desorption/ionization mass spectrometry imaging (MALDI-MSI) revealed significant increases of ATP/AMP ratio and of a specific sphingomyelin species (SM(d18:1/16:0)) in glomeruli of DKD mice [40]. In vitro, addition of SM(d18:1/16:0) to mesangial cells activated glycolysis. Fumarate was identified as a factor that mediated Nox4-induced injury in DKD [41]. Podocyte-specific induction of Nox4 in vivo recapitulated DKD-induced glomerular injury, and metabolomic analysis demonstrated increased fumarate, which was reversed with Nox1/Nox4 inhibition. Fumarate could serve as a hypoxia-inducible factor (HIF) stabilizer, which could cause the activation of glycolysis and suppression of OXPHOS.

### 3.2. Toxic Metabolites of Glucose Metabolism

Four major pathways branching from glycolysis are known to produce toxic intermediate metabolites: the polyol pathway, the hexosamine pathway, the advanced glycation end-products (AGEs) pathway and the protein kinase C (PKC) pathway [12] (Figure 3). Inhibition of each one of these pathways ameliorates hyperglycemia-induced injury in preclinical models [42,43,44,45]. Notably, the activation of any of these pathways can be instantly reversed with the restoration of euglycemia.

In particular, levels of 3-deoxyglucosone (3DG) and methylglyoxal (MGO), members of reactive carbonyl species (RCS) which are produced by the degradation of glyceraldehyde in the AGE pathway, are increased with the activation of glycolysis. RCS have highly reactive carbonyl groups and can modify protein and DNA. Proteins glycated by RCS leads to formation of AGEs. Extracellular AGEs can increase the crosslinking of matrices, leading to arterial stiffening [46]. AGEs can also bind to receptors for AGE (RAGE) and transduce various signals into cells, including nuclear factor-κB (NF-κB) activation leading to ROS formation, inflammation and fibrosis [47].

## 4. Non-Glucose-Induced Mitochondrial Dysfunction in DKD

Although hyperglycemia is a key factor in the development of DKD, other factors involved in DKD can also contribute to mitochondrial dysfunction. Dyslipidemia and lipid overload is a common complication in DKD, and hypoxia in the tubulointerstium occurs regardless of the cause of chronic kidney disease (CKD) [48]. Endothelin-1 (Edn1) was first identified as a downstream factor of transforming growth factor-β (TGF-β) in a model of focal segmental glomerulosclerosis (FSGS) and shown to induce albuminuria via mtROS in glomerular endothelial cells. Later, the same signaling pathway was also found to be upregulated in DKD [49,50]. In this section, we discuss the roles of these factors in the development and progression of DKD in relation to mitochondrial dysfunction.

### 4.1. Lipotoxicity

In kidney biopsies of patients with DKD, extensive lipid droplet accumulation was observed by electron microscopy in glomerular endothelial cells, podocytes and tubular cells compared to healthy counterparts [51,52]. Genetic analysis of these samples revealed downregulation of fatty acid oxidation (FAO)-related genes including peroxisome proliferator-activated receptor (PPAR)-α, carnitine palmitoyltransferase 1 (CPT1), acyl-CoA oxidase, and L-FABP; upregulation of cholesterol receptors including low-density lipoprotein (LDL) receptors, oxidized LDL receptors, and acetylated LDL receptors; and downregulation of cholesterol-efflux-related genes including ATP-binding cassette transporter A1 (ABCA1), ATP-binding cassette transporter G1 (ABCG1), and apolipoprotein (APOE) [51]. While downregulation of FAO-related genes suggests a decrease in mitochondrial lipid metabolism as the cause of lipid accumulation, lipid accumulation itself can induce mitochondrial dysfunction. We previously showed that human podocytes treated with the serum of patients with DKD show increased tumor necrosis factor (TNF) expression and that local rather than systemic TNF causes free cholesterol accumulation and injury via the suppression of ABCA1 in podocytes [53,54]. Notably, ABCA1 suppression induced cardiolipin accumulation and peroxidation in mitochondria, sensitizing podocytes to injury [55]. ABCA1 overexpression or inhibition of cardiolipin peroxidation by elamipretide rescued podocyte injury in experimental DKD.

Tubular cells require large amounts of ATP for solute reabsorption and depend on FAO because fatty acids yield more ATP per gram than other energy sources [56]. In the early stage of diabetes, FAO is increased in accordance with increased FA flux, and ROS production is attributed to FAO, especially to electron leakage at the electron transfer flavoprotein that shuttles electrons from acyl-CoA dehydrogenases to coenzyme Q [57]. Nevertheless, FAO is eventually decreased in established diabetes as described earlier [51,58].

Tubular cells in patients with DKD are likely exposed to fatty acid-bound albumin, since dyslipidemia and proteinuria often accompany diabetes mellitus. Whereas albumin itself can cause tubular cell damage on its reabsorption, FA-bound albumin was shown to induce more severe tubular damage [59,60]. Cytotoxicity caused by FA or glycated albumin was shown to be mediated by the uptake via the protein cluster of differentiation 36/FA translocase (CD36/FAT) in the brush border in humans, in contrast to usual albumin reabsorption via a complex of megalin, cubilin and amnionless [61]. In addition to reabsorption, synthesis of FA is also upregulated [62,63]. FAs are esterified by long-chain acyl-CoA (LC-CoA) synthetase (ACSL), which is upregulated in both mouse db/db model and human DKD kidney samples [64,65]. LC-CoA is transferred to mitochondria via CPT1 and CPT2 to produce ATP via FAO and unmetabolized LC-CoA is cleaved or stored in lipid droplets to prevent lipotoxicity. When buffering ability is saturated, LC-CoA serves as an inhibitor of the Na^+^/H^+^ exchanger 1 (NHE1) and phosphatidylinositol 4,5-bisphosphate [PI(4,5)P2] binding to cause apoptosis in proximal tubular cells [66].

Genetic and pharmacological enhancement of FAO could represent a new therapeutic treatment strategy for patients with DKD. Transgenic expression of PCG1α and fenofibrate treatment ameliorated tubular cell apoptosis via restoration of CPTs and/or acyl-CoA oxidases [58,67].

### 4.2. Hypoxia

Tubulointerstitial hypoxia is known to be a final common pathway of CKD progression [48]. Regardless of the cause of the disease, hypoxia can occur due to decreased oxygen supply via blood flow because of impaired vasodilation, loss of vasculature, fibrosis, anemia, increased oxygen demand due to increased solute reabsorption, and inefficient ATP production because of mitochondrial uncoupling, which are the shared mechanisms in CKD.

Recently, the mechanism by which acute hypoxia causes mtROS production in the ETC has been revealed [68]. Acute hypoxia prompts a conformational shift in Complex I, leading to the activation of the Na^+^/Ca^2+^ exchanger in the inner mitochondrial membrane. Na^+^ imported into the matrix then interacts with phospholipids to reduce membrane fluidity, resulting in the inability of free ubiquinone to move between Complex II and Complex III. Thus, Complex III produces ROS.

Hypoxia had long been deemed a direct suppressor of OXPHOS because oxygen is indispensable as a receiver of electrons in the ETC. However, to decelerate OXPHOS acutely and directly, the oxygen concentration has to be as low as 0.3% [69]. In milder and prolonged hypoxia, HIF1α acts as a mediator to suppress OXPHOS and to increase glycolysis in order to prevent ROS production. HIF1α modifies ETC complexes and promotes metabolic shifts from aerobic OXPHOS to anaerobic glycolysis [69]. Nevertheless, in chronic hypoxia, these changes limit the mitochondrial ability to produce ATP and could induce ATP deficiency, possibly leading to cytotoxicity.

### 4.3. Endothelin-1 (Edn1)/Edn1 Receptor Type A (Endra) Signaling

Endothelin-1 (Edn1) was first characterized as a signaling molecule released by podocytes in TGF-β-induced FSGS [49]. Surprisingly, Edn1 causes mtROS, decreases reserve respiratory capacity and mtDNA damage in endothelial cells via Edn1 receptor type A (Ednra) activation, but this is not seen in podocytes. This pathway induces glycosaminoglycan degradation in the endothelial surface layer, leading to the loss of fenestration [70]. Interestingly, EDNRA activation in endothelial cells is also required for podocyte foot process effacement and apoptosis [49].

The similar role of mtROS and EDNRA in endothelial cells and Edn1 is also described in DKD [50]. Circulating Edn1 was increased in diabetic humans and mice [50,71]. Notably, Ednra was not detectable in the glomeruli of healthy human kidneys or DKD-resistant C57BL/6J mice, but was present in those of human DKD kidneys and diabetic DBA/2J mice. Ednra could be induced in high-glucose-treated podocytes, but these podocytes did not express Edn1, implicating other cell types or stimuli in this signaling cascade [50]. Thus, Edn1/Ednra signaling has a critical role in the reciprocal crosstalk between podocytes and endothelial cells via mitochondrial dysfunction both in FSGS and DKD.

Another vasoactive pathway, the renin–angiotensin–aldosterone system (RAAS), is also known to participate in CKD progression, including in DKD. Aside from the detrimental effects of increasing systemic and intraglomerular blood pressure, angiotensin II treatment was found to exacerbate mtROS production and mitochondrial fragmentation in podocytes both in vivo and in vitro, which can be reversed by mitoquinone, a mitochondria-targeted antioxidant [72].

## 5. Conclusions

Mitochondrial dysfunction plays a central role in the development and progression of DKD. Thus, targeting mitochondrial dysfunction in DKD could represent a novel therapeutic strategy for patients with DKD. However, as can be learned from studies investigating the beneficial effect of systemic antioxidant administration as a treatment for DKD, it seems that intervention in mitochondrial dysfunction has to be cell type- and context-specific. Another intriguing perspective is age- and sex-related difference in mitochondrial dysfunction in DKD. Although age is an important factor that affects mitochondrial function and the development and progression of DKD, not much is known about the exact mechanisms. In terms of sex, much difference is observed between males and females in the mitochondria of certain tissues, and estrogen and possibly testosterone can mediate renal mitochondrial bioenergetics [73]. Further investigations are needed to elucidate the exact mechanisms leading to mitochondrial dysfunction in DKD, to the extent that findings can be clinically applied to patients and change their prognoses.

## Figures and Tables

**Figure 1 biomolecules-12-00351-f001:**
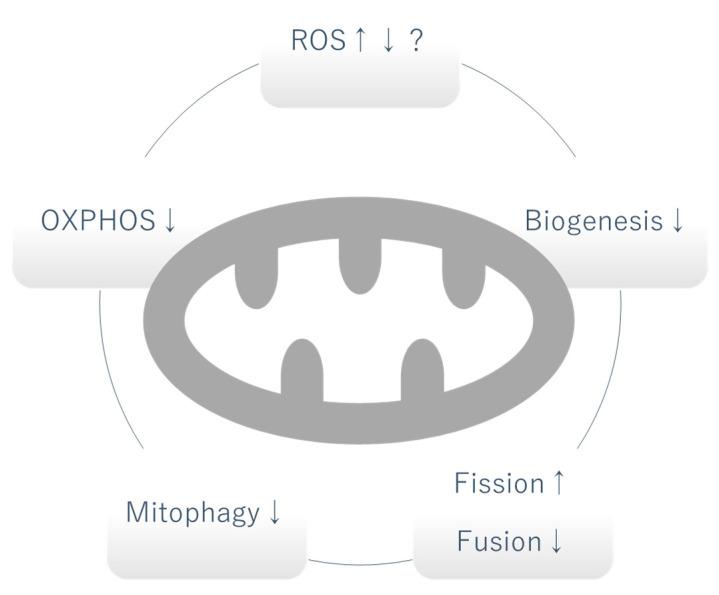
Mitochondrial dysfunction in DKD. Whether the level of mitochondrial ROS is increased or decreased is controversial and can vary depending on the stage of DKD. OXPHOS, mitophagy and biogenesis are generally decreased. Increased fission and decreased fusion causes fragmentation of mitochondria. OXPHOS: oxidative phosphorylation, ROS: reactive oxygen species, ↑: increased, ↓: decreased.

**Figure 2 biomolecules-12-00351-f002:**
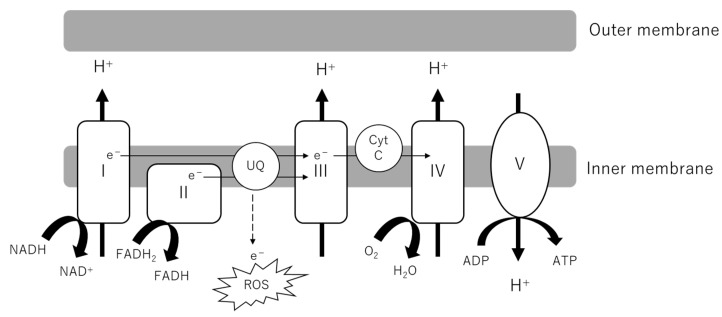
Electron transport chain (ETC) in mitochondrial inner membrane. NADH and FADH_2_ from the TCA cycle donate electrons to Complexes I and II. As electrons are transported through the ETC, a proton gradient is generated, which Complex V or ATP synthase couples to ATP synthesis. Electron leakage from the ETC causes the production of ROS. ADP, adenosine diphosphate; ATP, adenosine triphosphate; Cyt C, cytochrome complex; ROS, reactive oxygen species; UQ, ubiquinone; TCA cycle, tricarboxylic acid cycle.

**Figure 3 biomolecules-12-00351-f003:**
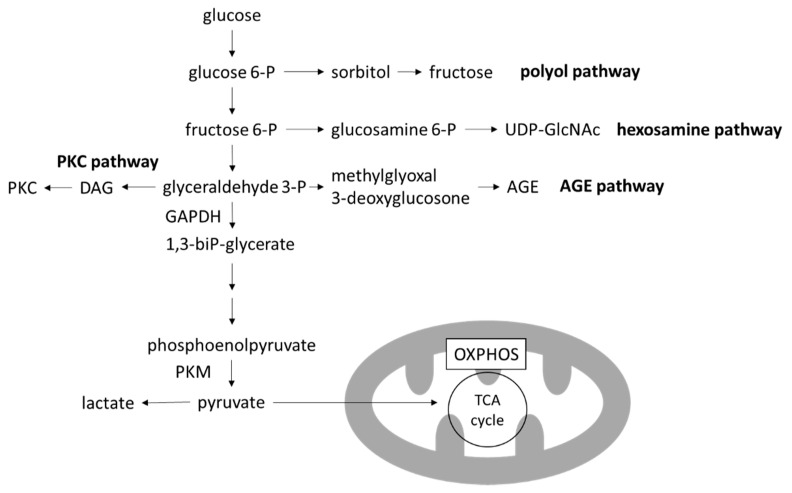
Glycolysis and branching pathways. In the high-glucose environment of DKD, glycolysis is increased and OXPHOS is decreased, leading to the accumulation of toxic metabolites produced in the branching pathways of glycolysis. AGE, advanced glycation end-product; DAG, diacylglycerol; GAPDH, glygeraldehyde-3-phosphate dehydrogenase; OXPHOS, oxidative phosphorylation; PKC, protein kinase C; PKM, pyruvate kinase M; TCA cycle, tricarboxylic acid cycle; UDP-GlcNAc, uridine diphosphate *N*-acetylglucosamine.

## Data Availability

Not applicable.

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
