# Peer review of "Glucose- and Non-Glucose-Induced Mitochondrial Dysfunction in Diabetic Kidney Disease"

_biomolecules, 2022, doi:10.3390/biom12030351_

Round 1
Reviewer 1 Report
The authors have completed an interesting review work highlighting important role of mitochondrial dysfunction in diabetic kidney disease, with both glucose- and non-glucose-induced factors. The work is useful and could represent a novel therapeutic strategy for patients with DKD. Congratulations to the authors for their great work. My suggestions are as follows:
Major changes
- A paragraph highlighting any evidence of age-wise and/or gender-wise effects observed on mitochondrial dysfunction in DKD (for both glucose- and non-glucose-induced factors in pre-clinical and clinical models) may improve the quality of presentation! Or a paragraph of study limitation highlighting all these above mentioned stuffs!
Minor changes
- In line 71, what are COQ5 and COX6A1? Coenzyme Q5 and Cytochrome C oxidase 6A1?
- In lines 116-117, you are re-defining PPARs and ERRs. These are already defined in line 112.
- In line 173, please fix “pro-tein” to “protein”.
- Please define “Akt” when using for the first time (line 176).
- In line 115, you used “NRF2”; and in line 178, you used “Nrf2”. Please fix it.
- I propose to add the citation for the sentence “Inhibition of each one of these pathways ameliorates hyperglycemia-induced injury in preclinical models” in lines 255-256.
- In lines 268 and 270, you used “OXYPHOS”, and in rest of the review work, you used “OXPHOS”. Please fix it.
- I propose to add the citation for the sentence “Genetic analysis of these samples revealed downregulation of fatty acid oxidation (FAO)-related genes including peroxisome proliferator-activated receptor (PPAR)-α, carnitine palmitoyltransferase 1 (CPT1), acyl-CoA oxidase, and L-FABP, upregulation of cholesterol receptors including low-density lipoprotein (LDL) receptors, oxidized LDL receptors, and acetylated LDL re- ceptors and downregulation of cholesterol efflux-related genes including ATP-binding cassette transporter A1 (ABCA1), ATP-binding cassette transporter G1 (ABCG1), and apolipoprotein (APOE).” in lines 286-293.
- In line 337, “Na+/Ca+” should be written as “Na+/Ca2+”. Please fix it.
- In lines 362 and 364, you used “Ednra”; and in line 360, you used “EDNRA”. Please fix it.
- In references 38, 47, and 49, the “year” should be “bold”.
Author Response
Thank you for your encouraging comment. We changed the draft according to your comments as below.
Major changes
- A paragraph highlighting any evidence of age-wise and/or gender-wise effects observed on mitochondrial dysfunction in DKD (for both glucose- and non-glucose-induced factors in pre-clinical and clinical models) may improve the quality of presentation! Or a paragraph of study limitation highlighting all these above mentioned stuffs!
→We added some sentences explaining these points in the paragraph 5. Conclusions.
Minor changes
- In line 71, what are COQ5 and COX6A1? Coenzyme Q5 and Cytochrome C oxidase 6A1?
→Yes, we added the terms accordingly.
- In lines 116-117, you are re-defining PPARs and ERRs. These are already defined in line 112.
→we erased the overlapping part.
- In line 173, please fix “pro-tein” to “protein”.
→Fixed.
- Please define “Akt” when using for the first time (line 176).
→Fixed.
- In line 115, you used “NRF2”; and in line 178, you used “Nrf2”. Please fix it.
→Fixed.
- I propose to add the citation for the sentence “Inhibition of each one of these pathways ameliorates hyperglycemia-induced injury in preclinical models” in lines 255-256.
→We added the citations.
- In lines 268 and 270, you used “OXYPHOS”, and in rest of the review work, you used “OXPHOS”. Please fix it.
→Fixed.
- I propose to add the citation for the sentence “Genetic analysis of these samples revealed downregulation of fatty acid oxidation (FAO)-related genes including peroxisome proliferator-activated receptor (PPAR)-α, carnitine palmitoyltransferase 1 (CPT1), acyl-CoA oxidase, and L-FABP, upregulation of cholesterol receptors including low-density lipoprotein (LDL) receptors, oxidized LDL receptors, and acetylated LDL re- ceptors and downregulation of cholesterol efflux-related genes including ATP-binding cassette transporter A1 (ABCA1), ATP-binding cassette transporter G1 (ABCG1), and apolipoprotein (APOE).” in lines 286-293.
→we added the citation.
- In line 337, “Na+/Ca+” should be written as “Na+/Ca2+”. Please fix it.
→Fixed.
- In lines 362 and 364, you used “Ednra”; and in line 360, you used “EDNRA”. Please fix it.
→Fixed.
- In references 38, 47, and 49, the “year” should be “bold”.
→Fixed.
Reviewer 2 Report
This review details the mechanisms in pathogenesis and progression of DKD via mitochondrial dysfunction based on many papers. This review is including an excellent discussion of the fact that there are conflicting studies regarding the sources and pathogenic role of ROS and changes in mtROS production. The distinction between the effects of hyperglycemia and mitochondrial dysfunction independent of blood glucose is also an excellent point of this paper. This paper deserves to be published because it is very useful for furthering the study of mitochondrial disorders and the development of DKD.
Author Response
Thank you so much for your compliment. We are very encouraged.
Reviewer 3 Report
I thank the authors for their thought-provoking review of the research regarding mitochondrial dysfunction in DKD. The discussion covers a range of important considerations and the key metabolic processes that are affected in DKD, alongside the discussion of both hyperglycaemia and dyslipidaemia in the aetiology of DKD. Each paragraph approaches the ideas logically and provides a great overview with appropriate examples from the literature. I enjoyed reading this review.
General:
A minor review of grammar is recommended.
Specific recommendations:
Paragraph 2.5 Mitophagy:
Cardiolipin is also an important ubiquitin-independent mitophagy stimulus that should be briefly included in this paragraph.
Paragraph 3.1 Warburg effect:
In the discussion of the Warburg effect it would be beneficial to also include the Crabtree effect as potentially underlying the glycolytic shift.
Optional recommendation:
Metformin and mitochondrial impacts:
Metformin is one of the standard therapies available for treating Type 2 Diabetes Mellitus. As there is evidence that it has an impact on glucose metabolism, hypoxia and mitophagy, it would be great to see a brief discussion on the clinical impact and importance of metformin to mitochondrial health in DKD as part of this review.
Author Response
Thank you for your kind and encouraging comments. We improved our draft according to your comments as below.
Specific recommendations:
Paragraph 2.5 Mitophagy:
Cardiolipin is also an important ubiquitin-independent mitophagy stimulus that should be briefly included in this paragraph.
→we added a part mentioning cardiolipin in lines 165-167.
Paragraph 3.1 Warburg effect:
In the discussion of the Warburg effect it would be beneficial to also include the Crabtree effect as potentially underlying the glycolytic shift.
→we added a part mentioning Crabtree effect in lines 218-220.
Optional recommendation:
Metformin and mitochondrial impacts:
Metformin is one of the standard therapies available for treating Type 2 Diabetes Mellitus. As there is evidence that it has an impact on glucose metabolism, hypoxia and mitophagy, it would be great to see a brief discussion on the clinical impact and importance of metformin to mitochondrial health in DKD as part of this review.
→Although we agree that the effect of Metformin on mitochondrial function is important, we decided to focus mainly on mitochondrial dysfunction and keep treatment outside the scope of this article.